# Post-COVID Kawasaki-like Multisystem Inflammatory Syndrome Complicated by Herpes Simplex Virus-1 in a Two-Year-Old Child

Emma L. Hodson [1,†], Iman Salem [1,*,†], Katherine E. Bradley [2,†], Chiamaka L. Okorie [2], Arthur Marka [1], Nigel D. Abraham [3], Nicole C. Pace [1,2,3], Alicia T. Dagrosa [1,2], Ryan C. Ratts [4,5] and Julianne A. Mann [1,2,3]

1   Department of Dermatology, Dartmouth-Hitchcock Medical Center, Lebanon, NH 03766, USA;
    emma_hodson@hotmail.com (E.L.H.); arthur.marka@hitchcock.org (A.M.);
    nicole.c.pace@hitchcock.org (N.C.P.); alicia.t.dagrosa@hitchcock.org (A.T.D.);
    julianne.a.mann@hitchcock.org (J.A.M.)
2   Geisel School of Medicine, Dartmouth College, Hanover, NH 03755, USA;
    chiamaka.l.okorie.med@dartmouth.edu (C.L.O.)
3   Department of Pediatrics, Dartmouth-Hitchcock Medical Center, Lebanon, NH 03766, USA;
    nigel.d.abraham@hitchcock.org
4   Department of Medicine, Section of Hospital Medicine, Dartmouth-Hitchcock Medical Center,
    Lebanon, NH 03766, USA; ryan.c.ratts@hitchcock.org
5   Department of Pediatrics, Section of Pediatric Hospital Medicine, Dartmouth-Hitchcock Medical Center,
    Lebanon, NH 03766, USA
*   Correspondence: iman.salem@hitchcock.org
†   These authors contributed equally to this work.

**Abstract:** Multisystem inflammatory syndrome in children (MIS-C) is a rare, systemic inflammation following severe acute respiratory syndrome coronavirus 2 (SARS-CoV-2) infection. We report a case of a 2-year-old male who presented with an exanthem and aberrant laboratory markers, mimicking Kawasaki disease but failing to meet the full diagnostic criteria. His course was further complicated by herpes Simplex Virus-1 (HSV-1) stomatitis.

**Keywords:** COVID-19; HSV-1; MIS-C; Kawasaki disease





## 1. Introduction

Most pediatric Coronavirus disease 2019 (COVID-19) cases are mild and usually asymptomatic compared to adult infection. However, in early 2020, numerous European reports described a constellation of multi-organ inflammatory symptoms following pediatric COVID infection, referred to as multisystem inflammatory syndrome (MIS) [1]. Since the condition is thought to be a post-infectious inflammatory process, treatment usually centers on immunosuppression [2]. Here, we describe the case of a two-year-old child who presented initially with a fever and a diffuse morbilliform rash. Laboratory investigation revealed markedly elevated inflammatory markers, pro-brain natriuretic peptide (proBNP), an altered coagulation profile, and impaired liver function. SARS-CoV-2 polymerase chain reaction (PCR) testing was negative, but nucleocapsid antibody testing was positive, suggesting prior COVID infection. The diagnosis of MIS-C versus Kawasaki disease (KD) was suggested. He initially improved with empiric treatment for both conditions but later developed a fever associated with severe oral mucositis. Herpes simplex virus 1 (HSV-1) PCR testing was positive. This report highlights the challenge of distinguishing MIS-C from KD. Additionally, this presentation illustrates the risk of opportunistic infections complicating the course of MIS-C syndrome and the importance of identifying symptoms of MIS-C to provide the most appropriate treatment for children.

## 2. Case Report

A previously healthy two-year-old male was admitted with a three-day history of fever reaching 102.2 F, vomiting, abdominal pain, malaise, and a rash first observed on the hands but quickly involving the rest of the body. He lived with his five siblings and parents, both of whom identify as individuals who smoke. His mother reported an older sibling having a sore throat one week prior. Streptococcal pharyngitis was considered based on symptoms, but rapid strep testing was negative. The mother had also developed mild upper respiratory symptoms. The patient's preadmission SARS-CoV-2 PCR test was negative. On examination, the patient appeared uncomfortable and ill with a diffuse, pink, morbilliform eruption involving the trunk and extremities with facial and flexural accentuation and acral edema (Figures 1 and 2). There was no blistering or sloughing. He had eye tearing without conjunctival injection. His lips and tongue appeared brightly erythematous (Figure 3). There was no cervical lymphadenopathy. Empiric intravenous antibiotics with Staphylococcus and Streptococcus coverage were administered, along with supportive fluids, after obtaining throat culture, skin swabs, and viral serology. The rash improved rapidly over two days. Blood testing was unremarkable except for the presence of SARS-CoV-2 nucleocapsid antibodies. In addition, labs revealed marked elevations of erythrocyte sedimentation rate (ESR), c-reactive protein (CRP), D-dimer, and liver enzymes, along with low albumin levels and platelet counts (Table 1).

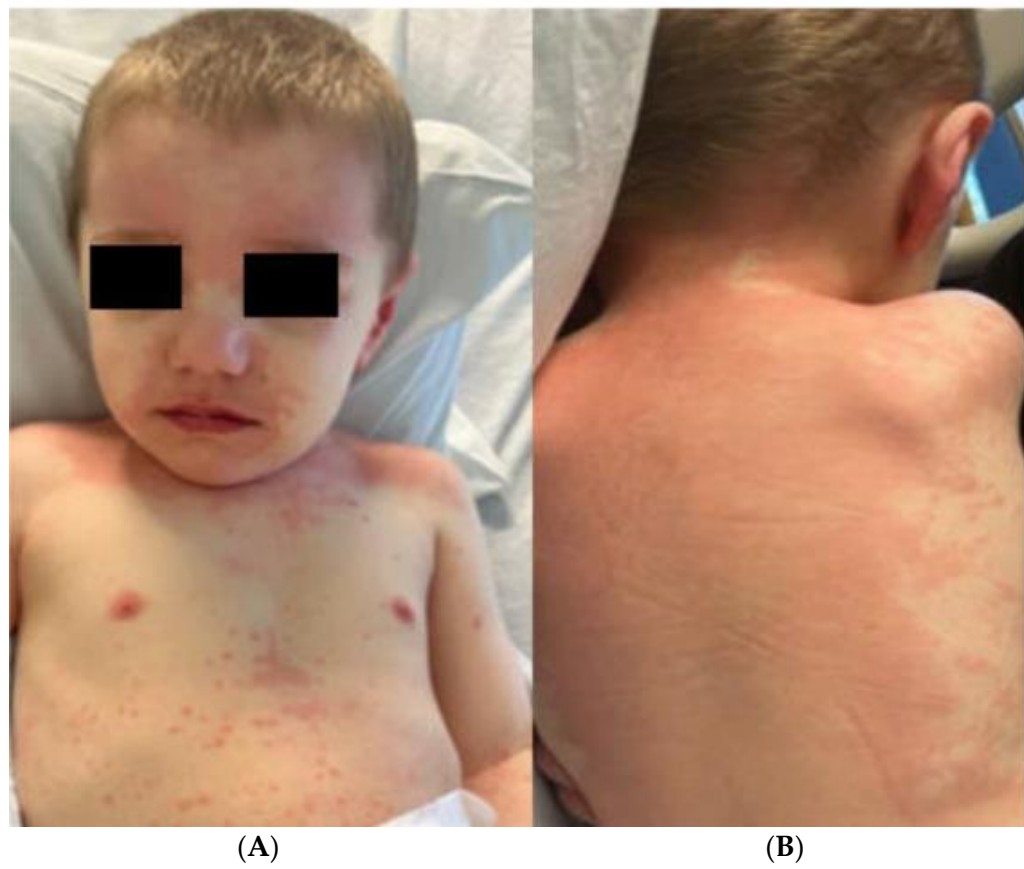

(**A**)　　　　　　　　　　　　　　　　　(**B**)

**Figure 1.** Multisystem inflammatory syndrome rash on face and trunk. Pink erythematous macules and edematous papules distributed over the abdomen and antecubital fossa. Rash coalesces to form patches and edematous plaques on periorificial regions, flexural folds (**A**) and back (**B**).

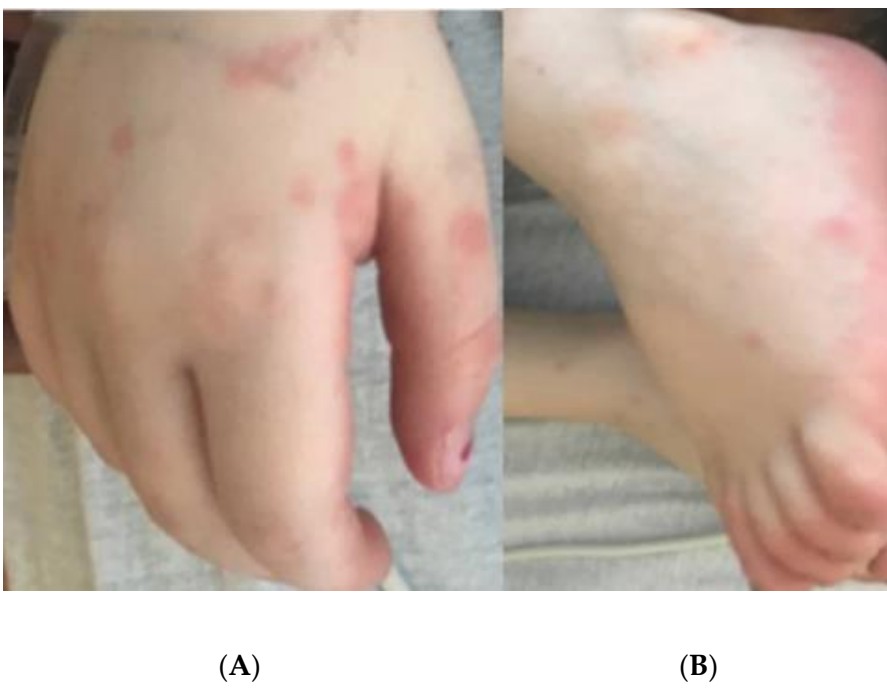

(A)  (B)

**Figure 2.** Multisystem inflammatory syndrome rash on extremities. Acral edema with erythematous pink macules and papules scattered over the dorsum of the hand and foot and coalescing into patches on the palm (**A**) and sole (**B**).

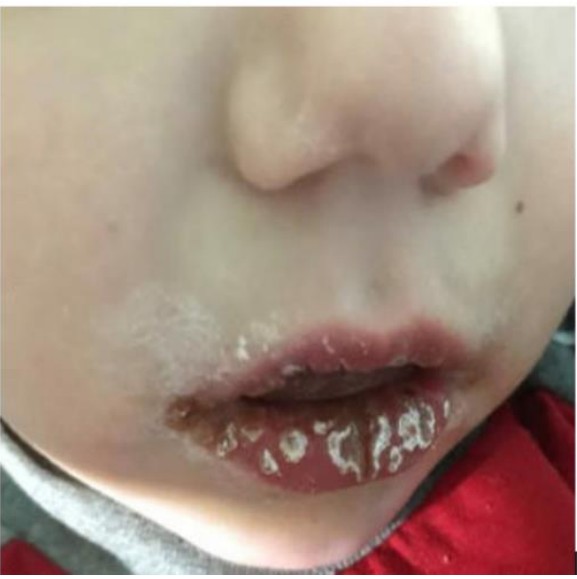

**Figure 3.** HSV-1 oral lesions. Erythematous and edematous vermillion lips with multiple crusted ulcerations.

A cardiac assessment on admission revealed a high proBNP without elevated troponin and a normal EKG. An echocardiogram demonstrated dilated coronary arteries without an aneurysm. Given the difficulty distinguishing MIS-C from KD and the cardiac risk associated with KD, pediatric cardiology advised treatment for KD: including two doses of intravenous immunoglobulin (IVIG 10% 35 g/350 mL total dose infused over 12 h), one-week of high-dose aspirin followed by a low-dose course, and IV methylprednisolone followed by oral prednisone. Following the first dose of IVIG, the patient required transfer to intensive care for hypoalbuminemia and fluid overload with associated edema, which required treatment with furosemide and albumin infusions. Repeated echocardiogram at

one week showed resolution of coronary arterial dilation. Although the patient initially improved, his condition subsequently worsened, with new findings of low-grade fever of 100.4 F and scalloped ulcerations with fibrinous exudate on the upper and lower vermillion lips. A complete blood count was significant for a dramatically worsening leukocytosis above 50,000 cells/mL, anemia, and thrombocytosis. The emergent oral lesions were found to be HSV-1 PCR positive. However, no testing was carried out for HSV-1 IgM antibodies. A patient's cousin had a history of cold sores, but the mother did not recall the patient having a personal history. The patient improved after receiving seven days of IV acyclovir.

**Table 1.** Lab results at hospitalization, after IVIG treatment, and at HSV-1 detection with PCR compared to normal pediatric range.

| Vitals and Laboratory Results | At Hospitalization | After Treatment | At HSV-1 Detection |
|---|---|---|---|
| Vitals | | | |
| Heart Rate | 142 | 99 | 122 |
| Respiratory Rate | 42 | 34 | 26 |
| Temperature (F°) | 104.6 | 99.3 | 102.2 |
| Lab (normal range) | | | |
| Ferritin (36–84) | 259 | 361 | 568 |
| C-reactive protein (<3) | 119.2 | 89.6 | 53.1 |
| Platelets (145–370 $\times 10^3$) | 199 | 121 | 799 |
| WBCs (5.5–15.5 $\times 10^3$) | 17 | 9.8 | 45.4 |
| Estimated sedimentation rate (2–34) | 48 | 3 | 25 |
| Alanine transaminase (0–34) | 299 | 62 | 31 |
| Aspartate aminotransferase (17–50) | 265 | 60 | 33 |
| Albumin (3.3–4.9) | 2.1 | 2.4 | 3.1 |
| Pro-BNP (≤124) | 17,990.00 | 15,790.00 | 8478.00 |
| D-dimer (215–780) | 5781.00 | 5668 | 2709.00 |
| Bilirubin (≤1) | 4.3 | 1.8 | 1.1 |

WBC, white blood cell; Pro-BNP, pro-brain natriuretic peptide.

## 3. Discussion

The most commonly used reliable test for diagnosing COVID-19 infection comprises the direct detection of one or more viral antigens using RT-PCR test. The viral genes amplified in this test include nucleocapsid, envelope, spike, and RNA-dependent RNA polymerase [3]. The sensitivity of PCR, however, was found to decrease two weeks after infection. In this case, the indirect measurement of host immune response can be helpful to indicate previous infection [3]. IgM and IgG ELISA can be positive even as early as the fourth day after symptom onset, and their levels continue to build up through the second and third week of illness [3]. A study including 140 participants revealed that during the first five to six days of infection, quantitative PCR had a higher positivity rate than IgM ELISA directed at nucleocapsid (NC) antigen, whereas IgM ELISA showed better detection rates after day 5.5 of illness [3]. In alignment with these findings, our patient's PCR testing was negative at presentation (one week after the initiation of his sore throat), while his serum was positive for SARS-CoV-2 NC antibodies.

Pediatric SARS-CoV-2 infection is generally mild, with only 1–5% of cases considered severe. Generally, the mild disease has been attributed to lower levels of angiotensin-converting enzyme 2 (ACE-2) expression in juvenile alveoli compared to adults [1]. Smoking is a risk factor for severe disease in adults as it triggers over-expression of the ACE-2 receptors, thereby increasing viral entry [4]. Our patient was exposed to second-hand smoke, possibly predisposing him to a severe inflammatory response following SARS-CoV-2 infection.

MIS-C is a potentially life-threatening post-COVID infection phenomenon that comprises, in addition to fever, the involvement of two or more organ systems and elevated inflammatory and hypercoagulability markers, such as CRP, ferritin, and D-dimer [1]. In the current case, the patient displayed a three-day history of fever (input number), abnor-

mal cutaneous, hepatic, and cardiac manifestations, and laboratory evidence of altered coagulation and inflammation (Table 1). Given the erythema and edema of the hands and feet and possible involvement of the oral mucosa in the setting of several days of high fever, KD was in the differential.

Kawasaki disease is a medium vessel vasculitis with a predilection for coronary arteries. It is the most common cause of acquired pediatric heart disease, frequently diagnosed in children younger than five years old [5]. The diagnosis of KD requires, in addition to a history of fever greater than 102.2 F/39 C for five or more days, at least four of the following features: bilateral bulbar conjunctivitis with extremely red eyes without thick discharge, cracked red lips with an exceedingly swollen red tongue, erythematous and edematous palms and soles with later peeling of skin on fingers and toes, cervical lymphadenopathy of 1.5 cm or more in diameter, and a diffuse maculopapular rash. The overlapping features of MIS-C and KD represent diagnostic and therapeutic challenges, such that MIS-C and KD could be considered on the spectrum of the same syndrome rather than two distinct conditions [6–8]. A meta-analysis by Tong et al., 2022 concluded that MIS-C has specific characteristics compared to KD, including older age of presentation, more frequent respiratory and gastrointestinal symptoms, higher levels of CRP, D-dimer, fibrinogen, pro-BNP, and AST, along with lower levels of platelets, lymphocytic count, albumin, and ESR [6]. However, KD cases were drawn from historical, rather than concurrent, cohorts [6]. To circumvent this limitation, a recent German study by Hufnagel et al., 2023, investigated the emergence of MIS-C and KD cases simultaneously using nationwide data collected from March 2020 to August 2021 [7]. Their results suggest that both conditions display more clinical and prognostic similarities than differences, therefore proposing the two conditions to be a syndrome continuum.

In our case (Table 2), although our patient's initial presentation did not meet the classic diagnostic criteria of KD (Figure 4), atypical KD was still considered, particularly in the setting of coronary artery dilation.

**Table 2.** Presenting signs and symptoms of the patient with overlapping features organized by disease group.

| MIS-C Features | Kawasaki Disease Features | HSV-1 Features |
|---|---|---|
| Fever; Gastrointestinal symptoms; maculo-papular rash; Extremity edema; Mucous membrane changes; Myocardial involvement with dilated coronary arteries without aneurysms and elevated pro-BNP; Hepatic involvement with elevated ALT, AST, and bilirubin | Recorded fever for 3 days (highest of 102.2 F); Morbilliform rash involving hands, feet, trunk, and extremities; Acral edema | Ulcers with crusting on upper and lower lip, vermillion border, and oral commissure |

While the duration of the fever was less than the five days required for a KD diagnosis, this could be secondary to the scheduled antipyretics the patient was receiving or not recognized by parents. Further, whereas the initial thrombocytopenia in our patient is more supportive of the diagnosis of MIS-C, KD can sometimes present with low platelets [9]. Additionally, our patient's symptoms of vomiting and abdominal pain, along with the elevation of his pro-BNP without change in troponin levels, support the diagnosis of MIS-C.

The pathogenesis of MIS-C still needs to be better understood. Despite the common association of MIS-C with exposure to SARS-CoV-2, a clear understanding of the pathophysiological link between the two has yet to be established. The predominance of data suggests MIS-C is a post-infectious inflammatory or autoimmune disorder rather than a severe infection [2]. This theory is primarily supported by the fact that only one-third of patients with MIS-C have positive PCR for SARS-CoV-2 at the time of diagnosis despite the detection of SARS-CoV-2 antibodies in most cases [9]. Multiple hypotheses are postulated to explain this observation. For example, the COVID spike protein encodes a high-affinity superantigen (SAg)-like sequence that exhibits strong binding to T-cell receptors, triggering amplified T-cell activation and proliferation with subsequent massive production of pro-inflammatory cytokines [10,11]. An alternative explanation

is found in the antibody-dependent enhancement (ADE) phenomenon. ADE describes an unconventional mechanism of viral entry independent of ACE2, where a complex of SARS-CoV-2 antigen and non-neutralizing virus-specific antibodies gains access to cells through binding to Fc receptors on the surface of immune cells [1]. Our patient's young age, positive COVID serology, and negative PCR for SARS-CoV-2 lend credibility to this theory. Another possibility is a persistent extra-pulmonary infection, especially isolating the virus from multiple organs, including the liver, heart, and gastrointestinal system [11]. Supporting this idea is a recent study by Thiriard et al. [12]. Their research revealed that MIS-C children produce higher titers of IgA and IgG antibodies. Remarkably, those IgG antibodies show higher functionality compared to pediatric patients with uncomplicated infection [12]. The authors provided an interesting explanation for their findings, where the exaggerated IgA immune response reflect a local gastro-intestinal mucosal inflammation possibly induced by a sustained SARS-CoV-2 gut infection leading to continuous release of viral antigens [12]. The abdominal symptoms in the current case could indeed back up this hypothesis although no IgA levels were tested in our patient.

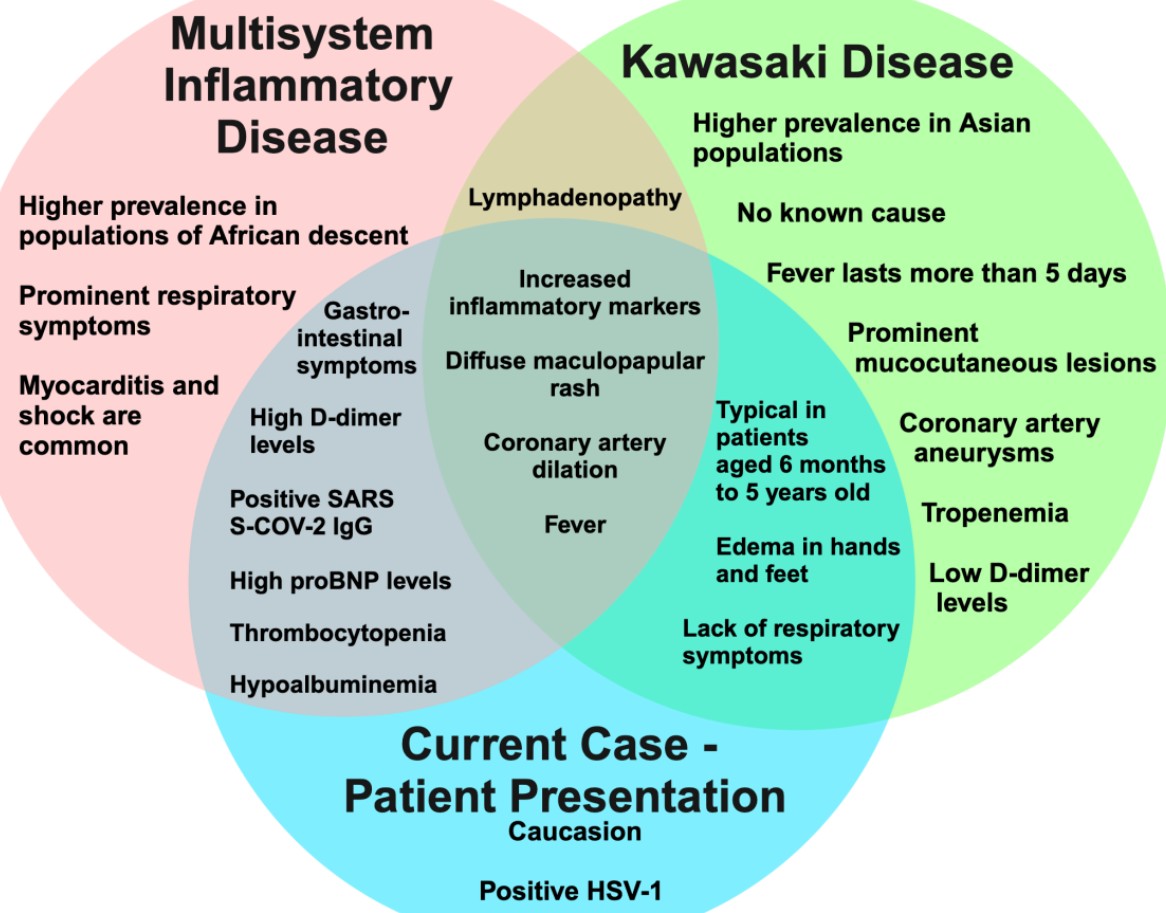

**Figure 4.** Similarities and differences between Multisystem inflammatory syndrome in children and Kawasaki disease compared to the current case.

The pathogenesis of MIS-C is thought to involve high levels of receptor-binding protein (RBD) antibodies [13]. And since both natural SARS-CoV-2 infection and BNT162b2 vaccination can expose the immune system to viral antigen eliciting RBD antibodies immune response, it is possible that MIS-C can be developed after either form of exposure [13]. Indeed, the frequency of MIS-C without evidence of SARS-CoV-2 after COVID-19 vaccination is very uncommon with an incidence rate lower than 1 per million vaccinated individuals

12–20 years [13]. Additionally, cases of MIS-C post vaccination are often very mild and rarely require ICU admission [13]. Current guidelines support high-dose corticosteroids, IVIG, and antithrombotic therapy as the first-line treatment of MIS-C [14]. This regimen overlaps with the treatment of KD except for the aspirin dose, which is typically high in KD and low in MIS-C. Despite the initial improvement observed in the first few days, the child's condition worsened with the development of HSV-1. The distinction between primary infection and reactivation was impossible in this case since IgM antibodies for HSV-1 were not tested. Further, because the patient previously received IVIG, there was no value in testing the patient for HSV-IgG.

Multiple theories could explain the development of this opportunistic infection in our patient. For example, SARS-CoV-2 could impair the IFN defense mechanism against HSV-1. Subsequently, the herpes virus can further suppress IFN-triggering genes leading to the amplification of innate immune suppression [15]. Another possibility stems from impaired adaptive immune response secondary to functional exhaustion of CD4+ and CD8+ T cells. Furthermore, adopting the superantigen hypothesis for developing a post-COVID infection, MIS-C could result in initial effector cell overstimulation, excessive cytokine production followed by T-cell exhaustion, and increased expression of inhibitory receptors [16]. Finally, the HSV-1 infection could be the reactivation of a latent infection secondary to immunosuppression from steroids and IVIG. Upon binding to their receptors, corticosteroids cease pro-inflammatory signaling pathways, inhibit transcription factors, and upregulate the expression of anti-inflammatory cytokines. This decreases T-cell effector subtype survival, differentiation, and function [17].

There is evidence of external risk factors further contributing to the development of several categories of skin eruptions and the exacerbation of pre-existing conditions such as psoriasis or HSV-1 following COVID-19 acute infections and vaccinations [18]. One predisposing risk factor is low vitamin D levels, which was shown to be predictive factor of a psoriatic flare post COVID-19 vaccination [18]. Furthermore, a recent study by Alshiyab et al., 2023, found a significant association between smoking and the development of skin eruptions during an acute COVID-19 infection [19]. Pertinent to our patient, as described above, second-hand smoke exposure is also associated with COVID-19-related skin eruptions, but this usually occurs in the context of a severe COVID-19 infection. However, our patient presented with a post-COVID-19 autoimmune or inflammatory disorder and HSV-1 reactivation. This raises an interesting question about the role of smoke exposure in these post-COVID-19 conditions and warrants further research.

### 4. Conclusions

The last three years have taught providers across all medical specialties to look at presentations through a pandemic lens. With its wide range of cutaneous complications and associations, this case adds to the discussion of COVID-19 as a new "great imitator". When faced with a diffuse rash in a critically ill child, a negative SARS-CoV-2 PCR cannot exclude COVID-19 involvement, and antibody confirmation should always follow. Through this description, we encourage reporting similar cases of MIS-C to enhance the understanding of the immune mechanisms and risk factors of this severe reaction. Additionally, we wish to draw attention to a possible link between secondhand smoking and increased susceptibility to intense inflammatory responses to COVID-19 infection. Finally, we acknowledge the challenge of differentiating MIS-C from KD. Patients with MIS-C and KD appear to be prone to opportunistic infections such as HSV-1, either from iatrogenic immunosuppressive therapies or environmental exposures, sequelae of the infection itself, or abnormal immune responses. Differentiating HSV-1 infections from orolabial involvement of MIS-C or KD is imperative, as missing this diagnosis could have resulted in administering additional immunosuppressive treatment and worsening the infection.

**Author Contributions:** Conceptualization, E.L.H., I.S., K.E.B., C.L.O., A.M., N.D.A. and J.A.M.; Writing—original draft, E.L.H., I.S., C.L.O., A.M. and N.D.A.; Writing—review and editing, E.L.H.,

I.S., K.E.B., C.L.O., A.M., N.D.A., N.C.P., A.T.D., R.C.R. and J.A.M.; Supervision, N.C.P., A.T.D., R.C.R. and J.A.M. All authors have read and agreed to the published version of the manuscript.

**Funding:** This research received no external funding.

**Informed Consent Statement:** Since the patient in the current report is a minor, informed consent was obtained from his care giver, his mother, including permission to publish case-related photos.

**Conflicts of Interest:** The authors declare no conflict of interest.

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
