# Peer review of "Post-COVID Kawasaki-like Multisystem Inflammatory Syndrome Complicated by Herpes Simplex Virus-1 in a Two-Year-Old Child"

_dermato, doi:10.3390/dermato3030017_

Round 1

Reviewer 1 Report

This is a case report of a 2-year-old male who presented with an exanthem and elevated inflammatory markers diagnosed as MIS-C syndrome mimicking Kawasaki disease and his course was further complicated by HSV-1 stomatitis. A further discussion on Kawasaki and MIS-C syndrome as post-COVID entities followed.

The case-report is well written and I really enjoyed the images and the diagram. The discussion section was very good too. Also the conclusion points are excellent.

Minor concerns

1)     a comparison between SARS-CoV-2 PCR testing and nucleocapsid antibody testing in terms of COVID-19 diagnosis should be briefly mentioned

2)     a brief reference of the following new citation that focuses on the antibody response between the MIS-C children and the children with uncomplicated COVID-19:

Thiriard A, Meyer B, Eberhardt CS, et al. Antibody response in children with multisystem inflammatory syndrome related to COVID-19 (MIS-C) compared to children with uncomplicated COVID-19. Front Immunol. 2023;14:1107156. Published 2023 Mar 15. doi:10.3389/fimmu.2023.1107156

3)     also reports of MIS-C children cases after COVID-19 vaccinations should be added in the discussion section and a brief comparison between COVID-19 infection cases

4)     finally the role of external contributors in the development of post-COVID 19 and post- COVID 19 vaccination skin eruptions should be briefly  mentioned. You can use this citation:

Karampinis E, Goudouras G, Ntavari N, Bogdanos DP, Roussaki-Schulze AV, Zafiriou E. Serum vitamin D levels can be predictive of psoriasis flares up after COVID-19 vaccination: a retrospective case control study. Front Med (Lausanne). 2023;10:1203426. Published 2023 May 25. doi:10.3389/fmed.2023.1203426

Author Response

We thank the reviewer for taking the time to review our manuscript and for his valuable comments and suggestions. Below is the point-to-point response to the comments

  • a comparison between SARS-CoV-2 PCR testing and nucleocapsid antibody testing in terms of COVID-19 diagnosis should be briefly mentioned

Response: Thank you for the suggestion. A paragraph was added in the beginning of the discussion section (lines 97 through 109) to briefly discuss this comparison as it pertains to our case.

2)     a brief reference of the following new citation that focuses on the antibody response between the MIS-C children and the children with uncomplicated COVID-19:

Thiriard A, Meyer B, Eberhardt CS, et al. Antibody response in children with multisystem inflammatory syndrome related to COVID-19 (MIS-C) compared to children with uncomplicated COVID-19. Front Immunol. 2023;14:1107156. Published 2023 Mar 15. doi:10.3389/fimmu.2023.1107156

 Response: Thank you for the suggestion and providing this reference. A paragraph was added discussing this reference in lines 180 through 188.

  • also reports of MIS-C children cases after COVID-19 vaccinations should be added in the discussion section and a brief comparison between COVID-19 infection cases

Response: Thank you for the suggestion. A paragraph was added discussing this idea in lines 189 through 196.

4)     finally the role of external contributors in the development of post-COVID 19 and post- COVID 19 vaccination skin eruptions should be briefly mentioned. You can use this citation:

Karampinis, E., Goudouras, G., Ntavari, N., Bogdanos, D.P., Roussaki-Schulze, A.V. and Zafiriou, E., 2023. Serum vitamin D levels can be predictive of psoriasis flares up after COVID-19 vaccination: a retrospective case control study. Frontiers in Medicine, 10, p.1203426.

Response: Thank you for the suggestion and providing this reference. A paragraph was added discussing this idea in lines 219 through 230.

Reviewer 2 Report

A case study of a 2-year-old male with symptoms and laboratory markers resembling Kawasaki disease but not meeting its full diagnostic criteria following SARS-CoV-2 infection, indicative of the rare MIS-C, further complicated by HSV-1 stomatitis.

The manuscript is interesting, but very messy: double spacing between words, capital letters in the middle of a sentence, no explained abbreviations, the font on the figure is so small that you can't read it.

There is no information about the patient consent form, which is particularly important as the manuscript contains photos of the child.

In abstract: HSV-1 should be explained.

COVID-19, ESR, CRP, IVIG should be explained.

Figure4: the font is too small, it can not be read.

Double spaces appear in the text and figures, and capital letters appear in the middle of a sentence. Please fix it.

Fig4: the title should be just: “Similarities and differences between Multisystem inflammatory syndrome in children and Kawasaki Disease Compared to Current Case.”

Table 1: Abbreviations are not explained.

Dose of IVIG.

The height of the fever, in the case of KD is important.

The manuscript would benefit if the authors added another table with all the symptoms, which are indeed described in the text, but in different parts.

Author Response

dermato-2410496 Authors' Response

Response to Reiviewer #2

We thank the reviewer for taking the time to review our manuscript and for providing valuable comments and suggestions.

1) The manuscript is interesting, but very messy: double spacing between words, capital letters in the middle of a sentence, no explained abbreviations, the font on the figure is so small that you can't read it.

Response: Thank you for the comment. We have addressed this issue throughout the manuscript. A clearer version of figure 4 was also provided.

2) There is no information about the patient consent form, which is particularly important as the manuscript contains photos of the child.

Response: Thank you for the comment. A statement was added at the end, lines 297 and 298, to address this point.

3) In abstract: HSV-1 should be explained. COVID-19, ESR, CRP, IVIG should be explained.

Response: Thank you for the comment. All the abbreviations have been explained at least once throughout the manuscript.

4) Figure4: the font is too small; it cannot be read.

Response: Thank you for the comment. A clearer version of figure 4 was also provided.

5) Double spaces appear in the text and figures, and capital letters appear in the middle of a sentence. Please fix it.

Response: Thank you for the comment. We have addressed this issue throughout the manuscript.

6) Fig4: the title should be just: “Similarities and differences between Multisystem inflammatory syndrome in children and Kawasaki Disease Compared to Current Case.”

Response: Thank you for the comment. This was corrected as suggested.

7) Table 1: Abbreviations are not explained.

Response: Thank you for the comment. Explanation for all the abbreviations was added.

8) Dose of IVIG.

Response: Thank you for the comment. The dose of IVIG was added in lines 83 and 84.

9) The height of the fever, in the case of KD is important.

Response: Thank you for the comment. The height of fever was added in lines 47 and 89.

10) The manuscript would benefit if the authors added another table with all the symptoms, which are indeed described in the text, but in different parts.

Response: Thank you for the suggestion. Table 2 was added to summarize the patient’s symptoms.

Round 2

Reviewer 2 Report

The authors addressed the comments properly. I have no further comments. Thank you.